# Osteosarcopenia in Patients with Chronic Obstructive Pulmonary Diseases: Which Pathophysiologic Implications for Rehabilitation?

**DOI:** 10.3390/ijerph192114314

**Published:** 2022-11-02

**Authors:** Lorenzo Lippi, Arianna Folli, Claudio Curci, Francesco D’Abrosca, Stefano Moalli, Kamal Mezian, Alessandro de Sire, Marco Invernizzi

**Affiliations:** 1Physical and Rehabilitative Medicine, Department of Health Sciences, University of Eastern Piedmont “A. Avogadro”, 28100 Novara, Italy; 2Dipartimento Attività Integrate Ricerca e Innovazione (DAIRI), Translational Medicine, Azienda Ospedaliera SS. Antonio e Biagio e Cesare Arrigo, 15121 Alessandria, Italy; 3Physical Medicine and Rehabilitation Unit, Department of Neurosciences, ASST Carlo Poma, 46100 Mantova, Italy; 4Department of Rehabilitation Medicine, First Faculty of Medicine, Charles University and General University Hospital, 12800 Prague, Czech Republic; 5Physical and Rehabilitative Medicine Unit, Department of Medical and Surgical Sciences, University of Catanzaro “Magna Graecia”, Viale Europa, 88100 Catanzaro, Italy; 6Department of Rehabilitation and Sports Medicine, Second Faculty of Medicine, Charles University and University Hospital Motol, 15006 Prague, Czech Republic

**Keywords:** osteoporosis, sarcopenia, chronic obstructive pulmonary disease, pulmonary rehabilitation, physical exercise, dietary supplements, rehabilitation

## Abstract

Chronic obstructive pulmonary disease (COPD) is a burdensome condition affecting a growing number of people worldwide, frequently related to major comorbidities and functional impairment. In these patients, several factors might have a role in promoting both bone and muscle loss, including systemic inflammation, corticosteroid therapies, sedentary behaviours, deconditioning, malnutrition, smoking habits, and alcohol consumption. On the other hand, bone and muscle tissues share several linkages from functional, embryological, and biochemical points of view. Osteosarcopenia has been recently defined by the coexistence of osteoporosis and sarcopenia, but the precise mechanisms underpinning osteosarcopenia in patients with COPD are still unknown. In this scenario, a deeper understanding of the molecular basis of osteosarcopenia might guide clinicians in a personalized approach integrating skeletal muscle health with the pulmonary rehabilitation framework in COPD. Taken together, our results summarized the currently available evidence about the multilevel interactions between osteosarcopenia and COPD to pave the way for a comprehensive approach targeting the most common risk factors of these pathological conditions. Further studies are needed to clarify the role of modern clinical strategies and telemedicine solutions to optimize healthcare delivery in patients with COPD, including osteopenia, osteoporosis, and sarcopenia screening in these subjects.

## 1. Introduction

Chronic obstructive pulmonary disease (COPD) is a highly prevalent respiratory disease characterized by airflow limitations and disabling respiratory symptoms, including chronic cough, increased sputum production, and exertional dyspnoea, resulting in physical function impairment and reduced health-related quality of life (HR-QoL). To date, COPD is considered a critical issue in the clinical setting given that it is frequently related to major comorbidities that contribute to an increased risk of hospitalization and mortality, including cardiovascular diseases, lung cancer, gastroesophageal reflux, metabolic syndrome, anxiety, depression, sarcopenia, and osteoporosis [1].

In particular, patients with COPD might be more frequently affected by both sarcopenia and osteoporosis compared to the general population [2], and the prevalence of both these pathological conditions is constantly increasing considering the progressive aging of the general population [3].

Sarcopenia has been recently defined by the European Working Group on Sarcopenia in Older People 2 (EWGSOP2) as a reduction of skeletal muscle strength associated with low muscle quantity or quality [4]. It is a generalized and progressive skeletal muscle disorder that can increase the risk of falls, fractures, physical disability, and mortality.

In this scenario, decreased activity and deconditioning is a major issue leading to peripheral muscle dysfunction in patients with COPD. More in detail, muscle mass decreases, and oxygen cannot be used effectively, resulting in a vicious cycle of reduced exercise capacity. Beyond chronic inflammation and reduced physical activity, factors that decrease muscle strength and endurance in COPD patients include oxidative stress, inactivity, hypoxemia, hormone abnormality, lack of nutrients such as protein and vitamin D, and the use of corticosteroids [5].

On the other hand, osteoporosis is a systemic bone disease characterized by low bone density and microstructural changes, mainly at the trabecular level, that increase the risk of fragility fractures [6]. Regrettably, osteoporosis in COPD patients is often undiagnosed and underestimated, given that it is an asymptomatic disease until a fragility fracture occurs. At the same time, low physical activity and long-term bed rest contribute to osteoporosis development and increase the risk of fragility fractures and adverse health outcomes, like worsening lung function and quality of life and increasing hospitalization and mortality rates.

Considering the functional, embryological, and biochemical linkages between bone and muscle tissues, a growing literature has proposed the term osteosarcopenia to characterize a condition where osteoporosis and sarcopenia coexist, emphasizing the need for the comprehensive and simultaneous management of both conditions. More in particular, recent research has highlighted that bone and muscle share multilevel interactions in patients with chronic inflammatory diseases, with intriguing implications for a comprehensive approach targeting both these tissues [7,8,9].

In the context of multidimensional management of individuals with chronic respiratory diseases, pulmonary rehabilitation (PR) is recognized as a cornerstone of the comprehensive non-pharmacological treatment framework [10] with well-documented evidence supporting its positive effects in reducing fatigue and dyspnoea and improving exercise capacity and HRQOL. PR includes exercise training, education, nutritional supplementation, and psychosocial support. In recent years, a growing emphasis has been placed on the assessment and treatment of different comorbidities, as they can have a significant impact on intervention outcomes. On the other hand, recent research has emphasized the key role of physical exercises and rehabilitation in treating and preventing both sarcopenia and osteoporosis as key components of the comprehensive approach that targets the multilevel risk factors of both diseases.

In this scenario, a precise assessment of the pathological mechanism strictly linking COPD patients and patients with osteosarcopenia might represent a cornerstone in developing patient-centred therapeutical strategies targeting the most relevant risk factors for worsening outcomes in these patients. However, to date, no previous review has focused on the mechanisms underpinning osteosarcopenia in patients with COPD.

Therefore, the aim of this narrative review was to characterize the pathophysiological mechanisms promoting osteosarcopenia in patients with COPD, providing a broad overview about the evidence currently available in the literature to guide physicians in integrating musculoskeletal management in the comprehensive treatment framework of COPD patients.

## 2. COPD and Osteosarcopenia: A Negative Bond for Physical Frailty

To date, COPD is considered a global burden affecting approximately 10.3% of people aged between 30 and 79 years old worldwide [11]. The World Health Organization (WHO) recently estimated that COPD is the third leading cause of death, with over 3 million deaths in 2019 [12].

In light of these data, there is an increasing interest in the current literature about comorbidities and clinical complications of COPD patients that might have a crucial impact on both physical and psychosocial health. In this scenario, frailty syndrome represents a detrimental issue affecting a growing number of people worldwide and is highly prevalent in elderly COPD patients [13]. In particular, the recent systematic review by Marangoni et al. [14] reported that frailty syndrome characterizes approximately 19% of COPD patients, with crucial implications for losing independence in the activity of daily living and even mortality [15]. In addition, frailty syndrome is commonly characterized by unintended weight loss, self-reported exhaustion, muscle weakness, slow walking speed, and low physical activity, worsening both physical and physiologic functions and significantly affecting HR-QoL and sanitary costs related to medical treatment and assistance [16]. Interestingly, growing evidence has highlighted the strict bond between frailty syndrome and sarcopenia [17] that affects about 15–55% of COPD patients [18]. To date, sarcopenia might also be considered an independent risk factor for fragility fractures due to diminished physical performance [4], with consequent crucial implications for balance impairment and an increased risk of falls [19]. On the other hand, several common risk factors shared by COPD, sarcopenia, and frailty have been identified, including alcohol consumption, cigarette smoking, low PA, poor diet, and older age [20,21].

In older adults, sarcopenia might reflect the impaired muscle quality related to several factors, including altered oxidative capacity, impaired muscle mitochondrial function, and abnormal systemic inflammation. All these pathophysiological pathways are frequent in COPD patients [22,23], and recent research has underlined that muscle–skeletal tissues in these patients might be affected by metabolic alterations related to deconditioning and the hypoxic state. These modifications promote the shifting towards a glycolytic muscle fibre distribution, with a reduced cross-sectional area and capillary density [24]. Moreover, it should be noted that a growing literature is currently highlighting a close link between muscle tissue and bone health [25]. In this scenario, about 38% of patients with COPD might be affected by osteoporosis, which is frequently associated with sarcopenia because both these pathological conditions share several risk factors [26] and molecular pathways [27,28]. Bone muscle crosstalk has been recently identified in regulating tropism of both these tissues through several molecular pathways, including IGF-1, sclerostin, myostatin, irisin, myonectin, mitochondrial dysfunction, and inflammatory cytokines [18,29,30]. In addition, the high prevalence of sarcopenia in patients with osteoporosis highlight this close connection, which goes beyond the sole mechanical stimuli and has a negative impact on both muscle and bone quality and density [20,31]. For a comprehensive view of the pathological pathways involved in osteoporosis and sarcopenia development in COPD patients, see Figure 1.

Taken together, this evidence underlines the need for the early identification of osteoporosis and sarcopenia because they often coexist as a new entity, which has recently been called osteosarcopenia. In this scenario, the precise identification of risk factors for impaired musculoskeletal health is mandatory to implement the tailored management of COPD patients and to reduce the detrimental consequences of fragility fractures in these patients.

## 3. Long-Term Corticosteroids Therapy in COPD Patients

To date, the most recent guidelines include inhaled corticosteroids (ICSs) or systemic corticosteroids in the pharmacological management of patients with COPD in order to improve pulmonary symptoms during both the maintenance phase and exacerbations [32]. In this scenario, the recent literature has focused on the detrimental consequences of corticosteroid administration on bone mineral density (BMD) [33]. Despite these considerations, the molecular pathways involved in corticosteroid-induced osteoporosis are far from being fully characterized. Interestingly, Oray et al. suggested that corticosteroid drugs might target osteoblast activity, affecting mainly bones with higher trabecular content and crucially affecting bone quality, mainly in the first months. In addition, it has been reported that up to 40% of patients undergoing long-term corticosteroid therapies have a higher risk of developing macroscopical or microscopical osteonecrosis, which might be strictly related to increased bone remodelling and bone resorption [34].

On the other hand, corticosteroid administration crucially impacts the whole skeletal muscle system, inducing not only bone health impairment but also muscular impairment, especially after long-term administration. It has been reported that muscle might be frequently affected by disrupting protein metabolism, promoting catabolic pathways and musculoskeletal changes in fibre architecture by increasing fast-twitch glycolytic type IIB muscle fibres and thus promoting muscle atrophy [35].

In addition, chronic inflammatory diseases with oxidative stress and impaired mitochondrial function represent another crucial factor in accelerating muscular atrophy [36], as recent works described the pivotal role of the ubiquitin-proteasome and the autophagy-lysosome machinery in muscular homeostasis, highlighting their role in muscle atrophy and sarcopenia development [37].

However, more studies should address the long-term effects of corticosteroids on muscle and bone tissue in COPD patients, as current evidence is still controversial.

### 3.1. Inhaled Corticosteroids

To date, it is widely accepted that the detrimental consequences of corticosteroid use on bone health might be related not only to long term glucocorticoid therapies, but even a single dose of glucocorticoids might increase fracture risk [38,39,40,41].

However, ICS administration cannot be compared to systemic administration [32]. In particular, the effects of ICS have been deeply studied in COPD patients, highlighting conflicting results. Although ICS might negatively affect BMD, the effects on fracture risk are controversial [26,42]. Interestingly, a recent narrative review [32] summarized the results of two meta-analyses and one Cochrane Collaboration group systematic review. The first two meta-analyses [43,44] reported an increased likelihood of bone fracture in patients receiving long-term ICS. On the other hand, the Cochrane review did not find a significant correlation with bone health in a 3-year follow-up period [45]. Hence, data about ICS are not conclusive despite strong evidence currently supporting the higher risk of fragility fractures in patients receiving systemic corticosteroids [32]. Moreover, to date, no studies have assessed the association between ICS and sarcopenia [46]. In light of these considerations, the comprehensive management of COPD patients should address other critical issues related to long term corticosteroid therapies, including diabetes or sarcopenia, that might have crucial effects on the risk of falls and fractures [5,47]. Therefore, clinicians should be aware of the systemic consequences of corticosteroid therapies in COPD patients, not to reduce the prescription of a life-saving intervention but to tailor effective primary prevention strategies to improve bone health in these patients.

### 3.2. Systemic Corticosteroids

In accordance with the GOLD ABCD tool [48], COPD patients belonging to groups B and D might complain of frequent exacerbations, which are strictly connected to increased risk of complications and worsening of the health status [49]. Moreover, the frequency of exacerbations can be linked to COPD severity, with exacerbation rates ranging between 0.85 and 2.00 per year in the beginning and advanced stages, respectively [50].

In this scenario, data in the literature have historically supported the effectiveness of systemic corticosteroids in COPD exacerbation, relieving pulmonary symptoms and improving forced expiratory volume (FEV1) and length of hospitalization [51,52]. However, systemic corticosteroids have been proven not to decrease COPD mortality rates (OR = 1.00, 95% CI 0.6 to 1.66). In addition, systemic oral or intravenous corticosteroids might be related to several adverse events, including transient hyperglycaemia (OR 2.79, 95% CI 1.86 to 4.19) [50], secondary infections [53], and mood and behavioural changes. On the other hand, osteoporosis has been described as a secondary effect of long-term administration of systemic corticosteroids, crucially interacting with bone metabolism and shifting the balance between bone deposition and bone reabsorption [51].

According to a systematic review by Chen et al., there are few data on the use of systemic corticosteroids and osteoporosis development in these patients [26]. Moreover, some findings are controversial, as some studies such as Hattioli et al. reported osteoporosis as a common complication in COPD in many elderly patients who did not undergo systemic corticosteroids [54]. On the other hand, a cross-sectional study highlighted that systemic steroid users were twice as likely to have one or more vertebral fractures when compared to non-steroid users, with an age-adjusted OR of 1.80 (95% CI, 1.08 to 3.07) [55]. Furthermore, many other studies did report a reduction of BMD in patients treated with oral corticosteroids, and the amount of BMD reduction seems to be related to the administered dose of corticosteroids [39].

Sarcopenia is a common feature for many COPD patients and is related to chronic inflammation, increased oxidative stress, hypoxemia, hormone abnormalities, and deficits of nutrients such as protein and vitamin D, but also to systemic corticosteroid usage [5]. In particular, the molecular pathway of the musculoskeletal functionality impairment caused by chronic corticosteroid administration apparently involves decreased mitochondrial enzymatic activity, which boosts oxidative damage. These data are in line with rising evidence on mitochondrial alterations in terms of lower mitochondrial bioenergetic, oxidative, and antioxidant capacity that leads to inflammatory response dysregulation and increased oxidative stress in aging subjects [23]. Current evidence is focusing on the role of mitochondrial dysfunction in chronic respiratory diseases, including COPD, in boosting oxidative stress, inflammation, apoptosis, senescence, and metabolic reprogramming [56] (see Figure 1 for further details).

In light of these considerations, current clinical good practice suggests that short-duration treatments with systemic corticosteroids could be effective in reducing the adverse effects at the bone and muscle levels [57].

## 4. Systemic Inflammation and Osteosarcopenia in COPD Patients

Chronic systemic inflammation is a key component of COPD affecting not only molecular pathways directly involved in the aetiology of the disease but also the severity, functional outcomes, and risks of complications. Interestingly, recent reports emphasized the strict link between COPD, systemic inflammation, and unhealthy lifestyle behaviour that characterize COPD patients (i.e., smoking and low physical activity levels) [58,59]. Moreover, a growing literature has reported a strict association between abnormal inflammation response and COPD pathogenesis. In particular, a recent review [60] highlighted that several associated cytokines might have a role in COPD development, including interleukin (IL)-1α, IL-4, IL-6, IL-7, IL-8, IL-10, and IL-12, and tumour necrosis factor (TNF)-α. Cytokine dysregulation may increase immune cell engagement, oxidative stress, and fibroblast stimulation, with detrimental consequences on both COPD progression and worse clinical outcomes [60]. As a result, it is not surprising that patients with COPD have higher blood levels of C-reactive protein (CRP), fibrinogen, and leucocytes [61].

Altogether, these factors may be considered as the basis of a vicious circle, promoting the development of cardiovascular diseases, cachexia, sarcopenia, and osteoporosis in COPD patients. In particular, bone metabolism might be crucially affected by pro-inflammatory cytokines interacting with several pathways involved in bone remodelling. In particular, recent research has underlined that TNF-α, IL-1, IL-6, and IL-17 are the most important cytokines triggering inflammatory bone loss [62]. In addition, proinflammatory cytokines negatively affect receptor activator of nuclear factor-κB ligand (RANKL) and macrophage colony-stimulating factor (M-CSF), with the consequent upregulation of osteoclast differentiation and function-promoting bone resorption [63]. On the other hand, systemic inflammation is a widely documented risk factor for sarcopenia, promoting imbalances between anabolic and catabolic pathways and resulting in muscle wasting. In this context, systemic inflammation is often linked to age-related mitochondrial impairment with the consequent increase in reactive oxygen species (ROS), and unbalanced nitric oxide synthase (NOS) [23,64]. Interestingly, it has been proposed that oxidative stress might promote protein synthesis impairment, inducing lipofuscin deposition and inadequate proteolysis [65]. Moreover, a growing literature has underlined a strong correlation among high CRP levels, IL-6, TNF-α, skeletal maturation, and muscle wasting [66,67].

Taken together, these findings showed that several cytokines are involved in both COPD, osteoporosis, and sarcopenia development, with intriguing implications in the pathophysiological molecular pathways linking these three disabling conditions.

Although the biological mechanisms underpinning osteoporosis and sarcopenia development in COPD patients have not been fully characterized yet, a deep understanding of molecular pathways involved in inflammation-related musculoskeletal impairment might represent a key target to implement a personalized therapeutic approach to COPD patients.

## 5. Physical Activity Level: Musculoskeletal Metabolism Consequences

Compared to age-matched healthy populations or patients with other chronic diseases, COPD patients are commonly less active. In addition, the level of physical activity in these patients decreases considerably over time, and more rapidly compared to non-COPD subjects, further increasing the risk of hospital admission and mortality [68]. The prevalent sedentary lifestyle and, eventually, the prolonged bed rest during exacerbation might promote muscle wasting and bone loss [69,70].

Worsening of airflow obstruction in COPD patients leads to hyperinflation and increased dyspnoea, with a negative impact on exercise tolerance and the level of physical activity. On the other hand, a sustained reduced level of physical activity might accelerate the progression of exercise intolerance and muscle depletion, with the consequent promotion of both sarcopenia and osteoporosis. Moreover, the disuse of the skeletal muscle system can be related to specific adaptations, including a reduction of type I fibres and lower oxidative enzyme activity and muscle capillaries, and even muscle fibre atrophy and a switch from low to fast fibres. These biological changes result in macroscopical reduction of muscle strength, muscle atrophy, and endurance impairment that further limits mobility and enhances a vicious cycle [70,71].

Although muscle atrophy is the result of accelerated proteolysis and decreased synthesis, muscle disuse also has a key role in the morphologic and metabolic changes at the skeletal muscle level in COPD patients [70]. Moreover, the reduction in physical activity levels is one of the most important cause of bone loss due to the increased bone resorption and reduced bone formation, a pathological mechanism similar to the muscle modifications observed after disuse [69].

In accordance with Frost’s mechanostat theory, the cross-regulation of both muscle mass and bone mass is promoted by mechanical forces directed on both tissues [72]. In particular, mechanical stimuli can trigger bone anabolism, enhancing mechanical resistance proprieties in order to prevent damages caused by increased mechanical load. On the other hand, the lack of mechanical stimuli can lead to bone resorption. From a biomechanical perspective, a decrease in muscle mass and muscle strength might reduce the mechanical load on the bone, leading to bone loss. These mechanisms might partly explain the close link between these two tissues and partly justify the high prevalence of sarcopenia and osteoporosis that coexist in several chronic pathological conditions. Nevertheless, muscle atrophy cannot completely explain osteoporosis, nor does the reduction in bone mass explain sarcopenia [73].

In this scenario, the concept of “bone–muscle crosstalk” includes several autocrine, paracrine, and endocrine pathways, where a mechanical signal produced by muscle contraction transmits an anabolic stimulus to nearby bones, but both bones and muscles play important roles also as secretory organs that share several biochemical interactions [74].

In recent years, growing evidence has shown that both muscle and bone can secrete a wide range of cytokines (myokines, such as myostatin, irisin, and IL-6; and osteokines, such as RANKL, OPG, and osteocalcin) to interact in an autocrine, paracrine, or endocrine manner [31] (Figure 1). Notably, exercise training is generally considered the best non-pharmacological method to improve the loss of muscle and bone mass in most disabling conditions, reducing risk factors for falls and improving balance. Moreover, it is also currently the most effective non-pharmacological intervention to improve skeletal muscle function in COPD patients with sarcopenia [75]. Exercise also plays a crucial role in regulating bone metabolism, growth, and development [76]. In general, therapeutic exercises are classified into aerobic (or endurance) training and strength (or resistance) training.

Resistance training and acute exercise were shown to attenuate the skeletal muscle’s expression of myostatin, which is a negative regulator of skeletal muscle growth and development and is linked to insulin sensitivity [77]. Moreover, it up-regulates a proportion of myogenin, thus improving muscle performance and counteracting skeletal muscle dysfunction in acute exacerbation of COPD [78].

On the other hand, aerobic training intensity seems to be directly related to increased plasma concentration of irisin, a newly discovered myokine that plays a positive role in regulating muscle mass by promoting myoblast differentiation [79]. COPD patients had lower serum levels of irisin secreted by skeletal myocytes compared with age-matched control subjects, and this was not associated with the potential exercise capacity, muscle volume, or pulmonary function [80]. Although the role of acute exercise on the release of irisin is still controversial, aerobic exercise training programs that are sustained over time, and regular long-term physical activity were shown to promote the upregulation of serum irisin levels, improving bioenergetic functions of skeletal muscle [80].

To date, resistance training has a widely documented effect in improving muscle tropism and strength and BMD. Nevertheless, the role of aerobic exercise in reducing bone loss is still controversial. While individualized moderate to high-intensity endurance training protocols associated with resistance training can have some effects on BMD [81], in older adults it is worth noting that low-intensity aerobic exercises, or simple regular walking, have been shown to be insufficient to increase muscle mass and BMD. Indeed, a sufficient stimulus is needed to trigger the mechanism of adaptation to exercise that is enhanced by feedback communication between bone and muscle; osteocalcin secreted by bone plays a crucial role in regulating muscle quantity, function, and movement adaptability by promoting nutrient absorption and catabolism of myofibers as well as the expression and secretion of IL-6 in skeletal muscles [31].

Taken together, regular physical activity associated with a targeted exercise training program with a primary goal of improving physical performance, strength, and muscle mass should be considered a non-invasive and non-pharmacological method to prevent and manage both osteoporosis and sarcopenia in patients with COPD.

## 6. Malnutrition, Skeletal Muscle Impairment, and Risk of Falling

Approximately 45% of COPD patients might have compromised nutritional status related to the imbalance between energy expenditure and energy intake. As a result, almost 50% of patients with severe COPD might experience unintentional weight loss [82]. Several factors might contribute to malnutrition, including dyspnoea, gastrointestinal problems, and anxiety, which contribute to poor appetite and reduced dietary intake [46]. In later stages of the disease, systemic inflammation, oxidative stress, and corticosteroid therapies might crucially affect energy expenditure, leading even to cachexia [83,84]. In this scenario, low body weight has been a widely recognized independent risk factor for low BMD, fragility fractures, and sarcopenia [21,38].

In this scenario, several metabolic pathways have been related to impaired nutrition in COPD patients, and most of them are not fully understood. However, it should be noted that malnutrition represents a crucial factor in worsening the patient’s clinical conditions, and a tailored multitarget intervention including nutritional management might have significant implications for a patient-centred approach.

As is well-known, aging may have detrimental consequences on metabolic efficiency and bioenergetic capacity, promoting chronic inflammatory states and several age-related disorders. In this scenario, particular attention should be paid to energy and protein intake, given the metabolism imbalance and the difficulties in triggering protein synthesis. Recent evidence has underlined that protein, essential amino acids, and aminoacidic precursor supplementation might have significant implications in skeletal muscle system anabolism [85,86,87]. Taken together, this evidence emphasized that a “rehabilitation nutrition” intervention might significantly interact with the biological pathways promoting both bone and muscle wasting in patients with COPD, with intriguing implications in strength and muscle mass [85,86], hence resulting in a lower risk of fall and fragility fractures [87].

Although macronutrient imbalance might have detrimental consequences on skeletal muscle system health, clinicians should be aware that other micronutrients might have an important role in a tailored approach to treat osteosarcopenia. Vitamin D plays a well-known role in calcium and phosphate intestinal absorption. However, elderly subjects might often be affected by hypovitaminosis D due to low dietary intake, low sunlight exposure, and reduced expression of vitamin D receptors [86]. However, it should be noted that hypovitaminosis D might be related to the increased serum levels of parathyroid hormone, resulting in higher bone turnover and an imbalance of osteoblast–osteoclast activity, with significant implications for osteopenia and osteoporosis development [88] (Figure 1).

Moreover, in recent years, an increasing interest has grown about the strict linking between hypovitaminosis D and sarcopenia. More in detail, some molecular pathways involved in sarcopenia pathophysiology have been linked with hypovitaminosis D. In this context, the lower expression of vitamin D receptors in skeletal muscles might interact with nucleo-cytoplasmatic transcriptional actions regulating muscle tropism, with consequent skeletal muscle loss [89].

On the other hand, several studies have underlined the role of vitamin D deficiency in the progression of pulmonary diseases, given the widely documented role of vitamin D in immune regulation and the effects of pulmonary inflammation in patients with COPD [90].

Despite these considerations, the aging process is also characterized by alterations in gut microbiota, which affect not only systemic inflammation but also nutrient bioavailability and gut absorption. In this scenario, supplementation with probiotics, prebiotics, and symbiotics seems to contribute to modulating metabolism, with also positive effects in terms of strength and improved muscle mass [86]. Moreover, supplementation of micronutrients and macronutrients shows a positive role in selecting health-beneficial bacterial flora [91], as rising evidence marks a central role of gut microbiota in triggering the inflammatory pathways underpinning chronic diseases such as COPD. Interestingly, a recent study focused on the role of probiotics on vitamin D and intestinal calcium absorption, highlighting their role in the prevention of hyperparathyroidism and BMD loss [92]. Lastly, recent research has emphasized the effects of several nutraceutical and nutritional interventions on inflammation and mitochondrial activity, suggesting potential implications of specific supplementation in targeting biomolecular pathways involved in skeletal muscle wasting [22].

Taken together, this evidence suggests that a comprehensive rehabilitation approach to osteosarcopenia might integrate a specific nutritional assessment in order to prevent or treat skeletal muscle wasting frequently characterizing older adults with COPD [85].

## 7. Smoke and Alcohol Consumption: Which Negative Influence?

Cigarette smoke is one of the most important risk factors for COPD [93]. It has been estimated that approximately 20–25% of smokers develop COPD due to the proinflammatory stimulus of the respiratory system and the abnormal inflammatory response [93,94]. Indeed, COPD patients frequently have a long history of smoking habits, sometimes even ongoing [69]. On the other hand, smoking is an independent risk factor for osteoporosis and fragility fractures, with a cumulative effect over time [95]. The mechanisms underpinning the negative bond between osteoporosis and smoking are not fully understood. Beyond the widely documented proinflammatory effects of smoke that might indirectly affect bone health, a direct effect of tobacco smoking has also been described [96]. In particular, nicotine might directly reduce both osteogenesis and angiogenesis, which have a key role in bone metabolism [97]. In addition, tobacco smoking has been related to altered intestinal permeability, and calcium absorption improves blood acidity, leading to higher bone turnover, and alterations in bone collagen synthesis [96]. Moreover, vitamin D deficiency and increased alcohol intake are commonly found in smokers, and oestrogen synthesis is also decreased [69,95].

In addition, smoking and nicotine intake is strictly linked to body weight and hormonal regulation of appetite, with both central and peripheral pathways involving catecholamines, leptin, peptide-YY, ghrelin, glucagon-like peptide-1, adiponectin, cholecystokinin, and orexin [98] (as depicted by Figure 1).

On the other hand, the high oxidative stress characterized by several oxygen and nitrogen free radicals might activate molecular inflammatory pathways such as mitogen-activated protein kinase (MAPK) and nuclear factor kappa-light-chain-enhancer of activated B cells (NF-kB). Therefore, abnormal inflammatory responses impact the skeletal muscle system, leading to muscular atrophy and sarcopenia [99].

On the other hand, COPD patients have a higher incidence of alcohol abuse, as a component of negative lifestyle behaviours frequently characterizing COPD patients [100]. In this context, the effect of alcohol consumption on BMD is widely documented, as many studies described alcohol as a risk factor in the development of osteopenia and osteoporosis [101], with bone loss caused by uncoupled bone remodelling, which has been linked to increased serum sclerostin and osteoprotegerin. Moreover, heavy drinking is considered an independent risk factor for risk of falling and fragility fractures [102].

In addition, high alcohol intake has been related to higher risk of malnutrition [103], with both impaired efficiency on macronutrient and micronutrient intake. More in detail, heavy drinkers might have impaired gut microbiomes and compromised epithelial integrity, hence developing inflammatory settings that contribute to the overgrowth of proteobacteria and thus unbalanced protein metabolism in a catabolic direction.

Moreover, chronic exposure of the organism to alcohol-induced toxic oxidative species results in variable deficiency in micronutrient intake (vitamin A, B1, B6, C, D, E, K, folate, calcium, magnesium, phosphate, iron, zinc, and selenium) [104].

On the other hand, to date, a direct correlation between alcohol intake and sarcopenia is not strongly supported by the current literature [105]. However, preclinical studies have suggested potential negative implications of alcohol abuse in protein metabolism [103]. In addition, heavy drinking has been associated with ectopic fat accumulation in skeletal muscle, with local detrimental inflammatory effects mediated by oxidative stress, triggering inflammation and disrupting glucose metabolism and consequently promoting muscular insulin resistance [106].

Altogether, these findings underline several mechanisms linking unhealthy lifestyle performance with osteosarcopenia in COPD patients. Thus, a comprehensive rehabilitation approach should include specific educational therapy aimed at discouraging unhealthy behaviours and enhancing the positive effects of rehabilitation on the skeletal muscle system health of people with COPD.

## 8. Osteosarcopenia Management in COPD Patients

To date, osteosarcopenia management in COPD patients still remains a challenge, despite several advances in recent years that have been made in understanding the mechanisms promoting skeletal muscle impairment [7]. In accordance with the evidence underlined in this narrative review, a comprehensive intervention is needed to target the multilevel interaction promoting osteosarcopenia in patients with COPD. In particular, physical activity promotion is a cornerstone in the management of both osteoporosis and sarcopenia [95,107]. On the other hand, several barriers might negatively affect physical activity in COPS patients, including hypoxia, dyspnoea, oxygen therapy, anaemia, and environmental and organizational issues [108]. Therefore, a tailored approach promoting physical activity is needed to better address the critical issue of sedentary behaviour in COPD patients [7].

In accordance, a personalized nutritional assessment and management should be considered in patients with COPD due to the high prevalence of malnutrition and the detrimental effects on skeletal muscle health [82]. In this context, attention should be paid to energy intake, vitamin D, calcium, proteins, amino acids, probiotics, and prebiotics in order to avoid catabolic states and the downregulation of systemic inflammation that frequently characterizes COPD patients [7,109].

In patients receiving corticosteroid therapies, the inhaled administration route should be preferred, and systemic corticosteroids should per administered for a shorter duration period [38]. On the other hand, long term corticosteroid therapies should be associated with antiresorptive drugs to counteract glucocorticoid-induced osteoporosis [110].

Moreover, pulmonary rehabilitation might have a key role in reducing the risk of exacerbation and complications related to pulmonary infections, leading to detrimental consequences in the skeletal muscle system and increased sanitary costs [111,112].

Lastly, much attention should be paid to lifestyle interventions aiming at reducing unhealthy lifestyle behaviours, including alcohol abuse, and smoking habits [102,113].

Taken together, this evidence underlines that several therapeutic approaches should be considered in patients with COPD that are at risk of osteosarcopenia. In order to better summarize the potential therapeutic strategies in patients with osteosarcopenia, Table 1 provides additional information about the potential therapeutic approaches to manage osteosarcopenia in patients with COPD and the shared risk factors between these diseases.

On the other hand, the first step for an effective treatment should be the early identification of the disorder, given the high impact on physical function and global health [101,114]. Moreover, the intervention should be individualized to the most important risk factors for the specific patients, including different healthcare professions, to better address the needed for a multidisciplinary and patient-centred approach to skeletal muscle disorders.

## 9. New Perspectives in Pulmonary Rehabilitation

COPD is a burdensome condition that needs to be addressed with a comprehensive multidisciplinary approach. Pulmonary rehabilitation (PR) is currently considered one of the key features of standard care in COPD [111], given the strong evidence supporting its value in improving health outcomes [112]. In light of the above considerations, growing evidence marks the crucial impact of a multidisciplinary and multidimensional approach [115], comprehensive of each healthcare professional involved in the clinical management of the patient. Indeed, addressing psychological and social issues in patients with COPD, assessing the nutritional assets, and promoting strategies for dismissing unhealthy lifestyle habits cover a crucial part in a comprehensive clinical approach.

However, it should be noted that the complexity of the action plan depicted above sets some intrinsic limitations in long-term clinical management, including barriers to services accessibility and program sustainability, given that COPD is a chronic condition.

In this context, the recent COVID-19 emergency accelerated the spread of technological solutions for healthcare delivery, particularly for frail people considering their needs due to this detrimental disease [116,117,118]. Indeed, the COVID pandemic stimulated the telerehabilitation with interesting advantages in overcoming barriers to accessibility and reducing sanitary costs [119,120,121,122].

Despite the intriguing implications of digital innovation and telerehabilitation in COPD patients, there is still a large gap in knowledge about the optimal strategies to promote this approach in clinical settings, and a comprehensive rehabilitation approach is still underestimated and underused [123].

To date, recent reports [13,124,125] have suggested that both hospital and community settings lack structured and sustainable rehabilitation strategies. In this context, telemonitoring could be a helpful tool, paving the way for delivering rehabilitation interventions and filling the gap between inpatient and outpatient settings [13]. Moreover, telerehabilitation might have positive implications for overcoming logistic difficulties, leading to the impossibility of reaching the hospital care units where the rehabilitation protocol is commonly delivered.

Osteoporosis and sarcopenia are common disorders frequently affecting patients with COPD, although they remain underestimated and undertreated, with detrimental consequences in terms of worsening health outcomes (higher risk of fall, hospitalization, incidence and severity of comorbidities, disability, and worse quality of life). As a matter of fact, modern clinical strategies should pivot on standard PR delivery but should also focus on early identification of osteopenia, osteoporosis, and sarcopenia [126]. Remarkably, telemedicine is shown to be a suitable tool for early detection of functional impairment, but there is still a lack of consensus about the standard methods to assess physical performance remotely [13].

Taken together, structured organizational models including digital innovation might have a role in close follow-up strategies for monitoring functional impairment and skeletal muscle consequences of COPD in order to enhance the multidisciplinary and interdisciplinary management of this disabling condition and to integrate osteosarcopenia management in the comprehensive approach to people with COPD.

## 10. Conclusions

COPD is a complex and highly disabling condition that affects several organs and systems, including bone and muscle tissue. Osteosarcopenia is a recent definition where osteoporosis and sarcopenia coexist and is widely observed as a common feature of several chronic conditions including aging and COPD. At present, a comprehensive approach to implement osteosarcopenia management in the pulmonary rehabilitation framework is still challenging, although several mechanisms have been identified to have a role in skeletal muscle system impairment in patients with COPD.

Despite this evidence, we are aware that the narrative review design did not allow us to provide data derived from statistical analysis underlining the link between osteoporosis and sarcopenia. However, this narrative review could provide a broad overview of the multilevel interactions promoting osteosarcopenia onset in patients with COPD. In this context, a comprehensive approach should have as a target the most common risk factors, including systemic inflammation, corticosteroid therapies, sedentary behaviours, deconditioning, malnutrition, smoking habits, and alcohol consumption. Lastly, a patient-centred approach must focus on the modifiable risk factors and implement them into sustainable and comprehensive management of COPD patients.

Further evidence is needed to better characterize the optimal therapeutic strategy for empowering COPD patient engagement in the management of both musculoskeletal health and COPD itself to overcome the barriers to participation of these frail patients.

## Figures and Tables

**Figure 1 ijerph-19-14314-f001:**
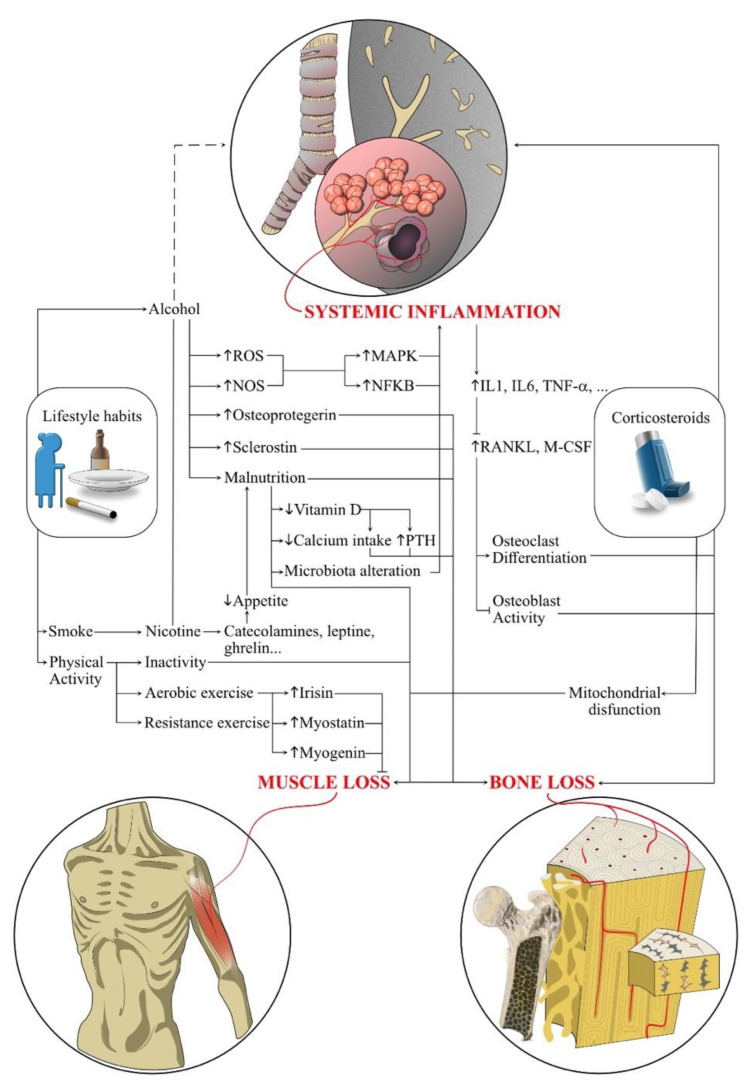
This figure lists the most important pathways involved in the pathophysiology of COPD, osteoporosis, and sarcopenia. *Abbreviations:* IL-(n): interleukin-n; MAPK: mitogen-activated protein-kinase; M-CSF: macrophage-colony stimulation factor; NFKB: nuclear factor kappa-light-chain-enhancer of activated B cells; NOS: nitric oxide synthase; PTH: parathyroid hormone; RANKL: receptor activator of nuclear factor kappa-Β ligand; ROS: reactive oxygen species; TNF-α: tumour necrosis factor-α.

**Table 1 ijerph-19-14314-t001:** Therapeutic approaches used in shared risk factors and complications of COPD and osteosarcopenia.

Area	Comment	Therapeutic Approach
Frailty syndrome	Highly prevalent in COPD patients and characterized by unintended weight loss, self-reported exhaustion, muscle weakness, slow walking speed, and low physical activity. It is independently linked to muscle and bone loss	Identification of risk factors for frailty syndrome and targeting of therapeutic strategies
Inhaled corticosteroids (ICS)	ICS are life-saving interventions, but their effects on BMD, sarcopenia, and fall and fracture risk are still controversial	Implement exacerbation prevention strategies to improve skeletal muscle health
Systemic corticosteroids	Systemic corticosteroids may cause long-term secondary osteoporosis and sarcopenia	Exacerbation prevention and short-duration treatments to reduce the adverse effects at bone and muscle levels
Systemic inflammation	Cytokine dysregulation might promote cardiovascular diseases, cachexia, sarcopenia, and osteoporosis	Inflammation-related intervention, including physical activity, physical exercise, pulmonary rehabilitation, nutritional approach, dietary supplements, and healthy lifestyle
Decreased levels of physical activity	Sedentary lifestyle and prolonged bed rest might promote muscle wasting and bone loss, as well as hyperinflation and increased dyspnoea due to airflow obstruction, decreased exercise tolerance	Personalized physical activity associated with a targeted exercise training program should be strongly encouraged
Compromised nutritional status	Low body weight and systemic inflammation might lead to cachexia, decreased BMD, fragility fractures, and sarcopenia	Nutritional assessment and management (“nutritional rehabilitation”), with particular attention to vitamin D, calcium, proteins, amino acids, probiotics, and prebiotics
Smoke and alcohol consumption	Unhealthy lifestyle behaviour (i.e., smoking and alcohol consumption) leads to proinflammatory stimulus	Educational therapy aimed at healthier lifestyle behaviours

*Abbreviations*: BMD: bone mineral density, COPD: chronic obstructive pulmonary disease.

## Data Availability

Not applicable.

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
