# Peer review of "Osteosarcopenia in Patients with Chronic Obstructive Pulmonary Diseases: Which Pathophysiologic Implications for Rehabilitation?"

_ijerph, 2022, doi:10.3390/ijerph192114314_

Round 1

Reviewer 1 Report

The review article titled "Osteosarcopenia in patients with chronic obstructive pulmonary  diseases: which pathophysiologic implications for rehabilitation?". In this authors want to summarize pathophysiological mechanisms promoting osteosarcopenia in patients with COPD. The review aims to helps the physicians to integrate the musculoskeletal management in comprehensive treatment framework of COPD patients.

Overall the manuscript is well written and extensive covering important aspects of the overall literature available and summarizing it well. 

Specific comments:

1. In line 108 incorporate third leading cause of death

 2. As the objective of the review is to help physicians in management and communicate overall similarity or commanalities  in the diseases a table containing common therapeutic approaches used in these complications is helpful.

3. Separate section highlighting therapeutic strategies is necessary.

Author Response

Dear Reviewer,

thank you for your letter and kind comments concerning our manuscript. We would like to express our sincere appreciation for your careful reviewing and invaluable comments which help us to further improve this paper.

The revisions based on your comments have been highlighted in the manuscript in yellow, and our detailed responses according to each revision are shown as followed.

Specific comments:

  1. In line 108 incorporate third leading cause of death

We would like to thank the Reviewer for the insightful comment. We improved the sentence following the Reviewer’s instructions.

  1. As the objective of the review is to help physicians in management and communicate overall similarity or commanalities in the diseases a table containing common therapeutic approaches used in these complications is helpful.

We would like to thank the Reviewer for the insightful comment. We included a Table summarizing the therapeutic approaches to counteract osteosarcopenia in patients with COPD.

  1. Separate section highlighting therapeutic strategies is necessary.

We would like to thank the Reviewer for the insightful comment. We improved the manuscript including a specific Section highlighting therapeutic strategies to counteract osteosarcopenia in patients with COPD, following the Reviewer’s instructions.

Reviewer 2 Report

The work is a narrative revision that wants to broaden the vision of the bronchopathic patient, and wants to frame the condition of osteosarcopenia in this pathology. the indications individually have been widely discussed in medicine and by different research groups, probably in this work we wanted to combine the various specialist contexts to lead them into a single vision in the specific treatment. the bibliographic references are good and the concepts expressed are clear. certainly not an experimental work that would certainly be clearer and more complete.
The argument is not original as indicated in the comment as the work is a narrative revision,
in practice, several works by other authors have been taken and have been included in a single article, wanting, as the only novelty, to combine the bpco with osteosarcopenia.
Also on the improvements I indicated that if it had been done as an experimental work, with specific research, it would have been clearly superior in terms of interest and specificity.
The conclusions of a review in this case are not derived from statistical studies of data from one population study but from inferences and experiences of other studies.
The tables are clear and relevant for the purpose of the specific explanations on the arguments.
The only new thing about this work, compared to previous studies, is the overview of the two diseases, BPCO and osteosarcopenia.

Author Response

Dear Reviewer,

thank you for your letter and kind comments concerning our manuscript. We would like to express our sincere appreciation for your careful reviewing and invaluable comments which help us to further improve this paper.

The revisions based on your comments have been highlighted in the Conclusions section in yellow, highlighting the limitation suggested by the Reviewer’s comment.
